# ATTENTION-BASED GUIDED STRUCTURED SPARSITY OF DEEP NEURAL NETWORKS

**Amirsina Torfi**
Virginia Tech
atorfi@vt.edu

**Rouzbeh A. Shirvani**
Howard University
rouzbeh.asgharishir@bison.howard.edu

## ABSTRACT

Network pruning is aimed at imposing sparsity in a neural network architecture by increasing the portion of zero-valued weights for reducing its size regarding energy-efficiency consideration and increasing evaluation speed. In most of the conducted research efforts, the sparsity is enforced for network pruning without any attention to the internal network characteristics such as unbalanced outputs of the neurons or more specifically the distribution of the weights and outputs of the neurons. That may cause severe accuracy drop due to uncontrolled sparsity. In this work, we propose an attention mechanism that simultaneously controls the sparsity intensity and supervised network pruning by keeping important information bottlenecks of the network to be active. On CIFAR-10, the proposed method outperforms the best baseline method by $6\%$ and reduced the accuracy drop by $2.6\times$ at the same level of sparsity.

## 1 INTRODUCTION

The main incentive behind model pruning is to impose sparsity by considerably reducing the number of effective parameters in a deep neural network while the accuracy drop is negligible (Han et al., 2015a; Denil et al., 2013). Different effective methods such as utilizing group lasso for learning sparse structure Yuan & Lin (2006), constrain the structure scale Liu et al. (2015), and regularizing multiple DNN structures known as Structured Sparsity Learning (SSL) (Wen et al., 2016) have been implemented for network pruning.

Unfortunately, there is a lack of addressing two issues for most of the conducted research efforts. First, pruning over-parameterized models with negligible accuracy drop, does not provide rigorous empirical proof for the effectiveness of the model since one can claim manually reducing the network size can generate relatively similar results (Zhu & Gupta, 2017). Second, imposing uncontrolled sparsity on under-parameterized baseline models may cause severe accuracy drop. Even if the network is over-parameterized, then imposing two much sparsity may cause the aforementioned issues.

In this work, we propose a controller mechanism for network pruning with the goal of (1) model compression for having few active parameters by enforcing group sparsity, (2) preventing the accuracy drop by controlling the sparsity of the network using an additional loss function by forcing a portion of the output neurons to stay alive in each layer of the network, and (3) capability of being incorporated for any layer type. Our source code is available online[1].

## 2 ATTENTION MECHANISM FOR GROUP SPARSE REGULARIZATION

The weights in a convolutional layer form a tensor as $W \in R^{C,[Width,Heigth],F}$ in which $C$ is the number of input-channel, $[Width, Heigth]$ is the spatial size of the kernel, and $F$ is the number of output filters (channels). In our proposed method, the objective is the minimization of the following loss function:

---

[1]https://github.com/astorfi/attention-guided-sparsity

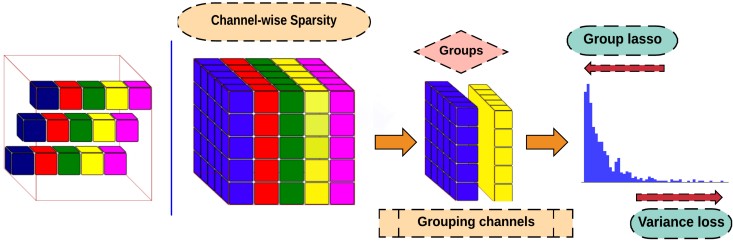

Figure 1: Channel-wise grouping and enforcing sparsity in addition to variance loss for each channel.

$$L(W) = L_{Softmax}(W) + \lambda_r.\ell_2(W) + \frac{1}{\sqrt{|G(W^l)|}}\{\lambda_{gs}.\sum_{l=1}^{N} L_{gs}(G(W^l)) + \lambda_{gv}.\sum_{l=1}^{N} L_{gv}^{-1}(G(W^l))\}$$

$$(1)$$

In the above equation, superscript $l$ indicates the layer index[2], $L_{Softmax}(W)$ is the Softmax loss, $\ell_2(W)$ is the $\ell_2$-regularization loss, and $L_{gs}$ and $L_{gv}$ are the group sparsity and group variance losses respectively. The value of $|G(W^l)|$ is essentially the number of channels for $l_{th}$ layer and $\lambda$ parameters are the hyper-parameter coefficients for the associated losses. The group sparsity regularization on a set of weights $w$ which are split into M groups can be shown as follows:

$$L_{gs} = \sum_{j=1}^{M} \sqrt{\sum_{i=1}^{|w^{(j)}|} (w_i^{(j)})^2}$$

$$(2)$$

in which $w^{(j)}$ is the $j_{th}$ group of partial weights in $w$ and $|w^{(j)}|$ is the number of weights in the associated group. Group sparsity has been employed due to its ability for deactivating neurons[3] by forcing the weights in a group to become zero[4] (Yuan & Lin, 2006; Meier et al., 2008). The loss function objective leverages group variance loss in addition to group sparsity loss to *force the distribution of the grouped weights to be skewed*. In another word, this attention mechanism, *simply emphasize on a high variance with a concentration around zero*. This will supervise the sparsity mechanism to deliberately keep a portion of grouped weights to be much larger than the majority of the groups in order to simultaneously sparse the architecture and prevent the accuracy drop. Intuitively, this operation forces a portion of channels to be active for transferring sufficient information through the channels in the whole architecture (information bottlenecks). The visualization of this reasoning is demonstrated in Fig. 1. So basically, in a convolutional layer, each group is all set of weights which forms an output channel. Equivalently, in a fully-connected layer, a group is the set of outgoing (ingoing) weights from a neuron. The group-variance is defined as below:

$$L_{gv} = \frac{1}{M}\sum_{j=1}^{M}\left(\sqrt{\sum_{i=1}^{|w^{(j)}|}(w_i^{(j)})^2} - \frac{1}{M}\sum_{k=1}^{M}\sqrt{\sum_{i=1}^{|w^{(k)}|}(w_i^{(k)})^2}\right)^2$$

$$(3)$$

In case of enforcing sparsity of output channels of convolutional layers, $W_{F_j}^{(i)}$ is the $j_{th}$ output channel of the $i_{th}$ layer, then the $L_{gs} = \sum_{j=1}^{N_{filters}}\sqrt{\sum(W_{F_j}^{(i)})^2}$ and so the formulation of $L_{gv}$ becomes straightforward. We call our method *Guided Structured Sparsity (GSS)* as it can be considered as an extension to SSL Wen et al. (2016) by having an attention mechanism using variational loss that is utilized for supervision of sparsity enforcement operation.

---

[2]In the range of [1:N] in case of having N layers.

[3]Channels in convolutional layer

[4]This effectively deactivate the neuron by canceling its output

## 3 EXPERIMENTAL RESULTS

We evaluated our proposed method on two databases: MNIST LeCun et al. (2010), CIFAR-10 Krizhevsky & Hinton (2009).In all our experiments we enforce the sparsity on both fc-layers (Using group sparsity for neurons inputs) and convolutional layers (Using channel-wise structured sparsity for eliminating unimportant filters). In the experiment on MNIST dataset, an architecture similar to LeNet LeCun et al. (1998) has been utilized as the baseline for investigation of our proposed method with no data augmentation. For experiments on CIFAR-10 dataset, we use the *ConvNet* provided by TensorFlow Abadi et al. (2015). The utilized baseline model contains two convolutional layers with *Local Response Normalization (LRN)* Krizhevsky et al. (2012) followed by two fully connected layers[5].

Table 1: Error rate for Different Methods on MNIST and CIFAR-10 at the same level of sparsity (90%).

| Method | Error(%) | |
|---|---|---|
| | **MNIST** | **CIFAR-10** |
| Baseline [*no sparsity*] | 0.93 | 15.51 |
| $\ell_1 - regularization$ | 3.16 | 24.84 |
| Network Pruning (Han et al., 2015b) | 2.67 | 23.12 |
| Sparsely-connected networks (Ardakani et al., 2016) | 1.91 | 17.12 |
| SSL (Wen et al., 2016) | 1.43 | 18.71 |
| **Guided Structured Sparsity [ours]** | **1.21** | **16.13** |

Table. 1 demonstrates the comparison results. It demonstrates that our method achieves less error rate compared to other methods.

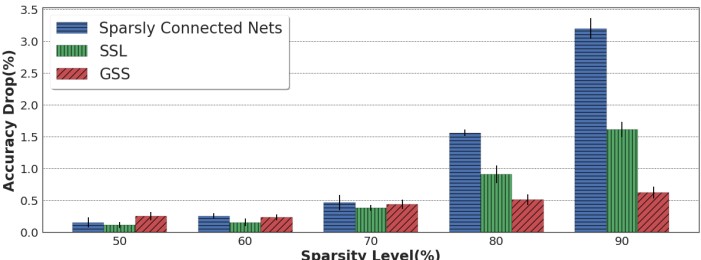

Figure 2: The results for experiments on CIFAR-10 dataset at different sparsity levels.

Fig. 2 depicts a comparison at different levels of sparsity. As it can be observed from the figure, our method demonstrates its superiority in higher levels of sparsity.

## 4 CONCLUSION

We have demonstrated that by utilization of the attention mechanism for sparsity supervision, a reduction of $2.6\times$ in accuracy drop has been obtained. Group sparse regularization has been employed on both convolutional and fully-connected layers for simultaneously imposing sparsity and demonstration of the adaptability of the proposed mechanism to both layer types. We anticipate greater superiority of our proposed method compared to the others by utilizing more complex models and evaluation on larger datasets. Besides, it is expected to show advancements in applications such as multi-modality fusion for which network pruning becomes of great importance due to the large number of weights and difficulties in learning a shared common feature space for all modalities (Ngiam et al., 2011; Zhao et al., 2015).

---

[5]Further details: `https://www.tensorflow.org/tutorials/deep_cnn`

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
