# OpenReview forum: "Attention-Based Guided Structured Sparsity of Deep Neural Networks"
_ICLR.cc/2018/Workshop — Reject_

### Official Review · AnonReviewer1 · 2018-03-09
**Nice contribution**

**Rating:** 7
**Confidence:** 3

**Review:**

This paper proposes an approach to enforce sparseness in deep neural networks in order to learn smaller models that maintain the accuracy of the full (over-parameterized) model. The proposed objective function is an extension of the structured sparsity loss proposed by Wen et al 2016. The extension is a group variance loss, which can be interpreted as applying an attention mechanism over the output channels’ weights. Under a 90% sparsity rate they obtain lower errors rates than Wen et al and other sparsity induction methods on MNIST and CIFAR-10.
This is a nice contribution as a workshop paper.

---

### Official Review · AnonReviewer2 · 2018-03-10
**Over complex regularization term, insufficient evaluation.**

**Rating:** 4
**Confidence:** 5

**Review:**

This paper propose to prune the network weights by using a group sparsity and a variance encouraging term to regularize the weights.
The text of paper is not very easy to read, for example, the name "attention mechanism" is not very well explained. I understand the idea mainly through the equation.

Pro:
This paper has proposed to add one more regularization term, namely the variance encourage term to the SSL method. It is experimentally shown that this regularization term is effective in pruning the network without losing much accuracy.

Con:
In my opinion, the regularizer itself is too complex to be an elegant method. It contains nested squares, square roots, and reciprocals, making the computation graph over complex.
The final loss function has three regularization terms each with a hyperparameter lambda to tune.
These are the two points I dislike about the idea of this paper.
Regarding experiments, the chosen architecture are too old to be convincing. One could simply argue that newer structures with better inductive bias could out perform the pruned network with the same number of parameter. As the authors also point out that manually reducing the network size could lead to similar results.

---

### Official Review · AnonReviewer3 · 2018-03-11
**A controller mechanism for network pruning is proposed based on previous works by further extending the group sparsity loss in SSL with group variance loss. Experiments demonstrate the robustness of proposed model.**

**Rating:** 5
**Confidence:** 4

**Review:**


Contributions:
1. The novel sparsity regularization is capable of being incorporated for any layer type (Conv & FC as given in paper)
2. Seen as an extension for SSL, only minor modifications are needed to implement this idea

Drawbacks:
1. Explanation on group variance loss in the Model Part is required with more details, and so is the last visualization in Fig.1
2. Experiments would be more convincing if large data-sets(ImageNet) and networks with complex architecture(ResNet) were included
3. When mentioning the “Information Bottleneck” in the paper, authors failed in giving related evidences

I’m open to change my opinion if more persuasive experiments are attached.

---

### Decision · Program_Chairs · 2018-03-20
**ICLR 2018 Workshop Acceptance Decision**

**Decision:**

Reject

**Comment:**

Based on the reviews, this paper has not been accepted for presentation at the ICLR workshop. However, the conversation and updates can continue to appear here on OpenReview.